# Structure-controllable growth of nitrogenated graphene quantum dots via solvent catalysis for selective C-N bond activation

Byung Joon Moon[1,2,7], Sang Jin Kim[1,7], Aram Lee[1,7], Yelin Oh[1], Seoung-Ki Lee[3], Sang Hyun Lee [4],
Tae-Wook Kim [5], Byung Hee Hong[2,6] & Sukang Bae [1✉]

Photophysical and photochemical properties of graphene quantum dots (GQDs) strongly depend on their morphological and chemical features. However, systematic and uniform manipulation of the chemical structures of GQDs remains challenging due to the difficulty in simultaneous control of competitive reactions, i.e., growth and doping, and the complicated post-purification processes. Here, we report an efficient and scalable production of chemically tailored N-doped GQDs (NGs) with high uniformity and crystallinity via a simple one-step solvent catalytic reaction for the thermolytic self-assembly of molecular precursors. We find that the graphitization of N-containing precursors during the formation of NGs can be modulated by intermolecular interaction with solvent molecules, the mechanism of wh ich is evidenced by theoretical calculations and various spectroscopic analyses. Given with the excellent visible-light photoresponse and photocatalytic activity of NGs, it is expected that the proposed approach will promote the practical utilization of GQDs for various applications in the near future.

[1] Functional Composite Materials Research Center, Korea Institute of Science and Technology, Wanju-gun, Jeollabuk-do 55324, Republic of Korea.
[2] Department of Chemistry, Seoul National University, Seoul 08826, Republic of Korea. [3] School of Materials Science and Engineering, Pusan National University, Busan 46241, Republic of Korea. [4] The School of Chemical Engineering, Chonnam National University, Gwangju 61186, Republic of Korea.
[5] Department of Flexible and Printable Electronics, Jeonbuk National University, Jeonju-si, Jeollabuk-do 54896, Republic of Korea. [6] Graphene Research Center, Advanced Institute of Convergence Technology, Seoul National University, Suwon-si, Gyeonggi-do 16229, Republic of Korea. [7] These authors contributed equally: Byung Joon Moon, Sang Jin Kim and Aram Lee. ✉email: sbae@kist.re.kr

Graphene quantum dots (GQDs) are a relatively new class of low-cost carbon-based fluorescent nanomaterials that have been recognized as great potential candidates for various research areas of photonics and biochemistry and applications for catalysis, energy conversion/storage, and sensing owing to their unique chemical, electrochemical, optical, and biological properties[1–3]. To be appreciated as a practical alternative to the previously used materials in numerous applications, the chemical composition or configuration of GQDs should be systematically manipulated, as these features are closely related to the biocompatibility, catalytic behaviour, chemical reactivity/ selectivity, and optical responses, e.g., emission dynamics, absorption/emission window, and photosensitivity, of the materials[4–7]. In an attempt to tailor the chemical features of GQDs, substitution of heteroatoms into the graphitic lattice and the introduction of specific chemical functional groups to the edge sites have been actively studied[4–6,8]. Recently, as a simple but efficient "bottom-up" method, one-pot pyrolysis using single or multiple organic precursors has been widely utilized to prepare heteroatom-doped GQDs with lateral sizes ranging from 1 to 10 nm due to the simplicity of the process[8–13]. Additionally, some reports have revealed the feasibility of obtaining edge-controlled GQDs by changing the types of solvents used during the formation processes[14,15].

Despite these early successes in the production of heteroatom-doped GQDs, there still exist many challenges in the optimization of fabrication processes, including the removal of acidic/basic intermediates and dopant residues[11,16–19], which necessarily involve time/cost-consuming post-treatments and hinder large-scale manufacturing. Furthermore, current techniques derived from solution-based one-pot methods are insufficient to allow simultaneous control of both the structural features (crystallinity) and doping characteristics (concentrations and configurations). Essentially, this limitation is caused by either the amorphous structure of dopants or the incompatibility with thermolysis because of the high volatility of dopants and the resulting increased internal pressure of the reaction system. Moreover, the uncontrollable kinetics of subsequent competitive reactions, such as growth of graphitic domains and additive (dopant)-induced side reactions that form molecular fluorophores, may impede an intuitive prediction of the synthesis[10,16,20,21].

Additionally, further development of such synthetic strategies towards diverse applications has been hindered by the absence of detailed investigations into the correlation between optical properties and the surface status of doped GQDs, which is passivated by the incomplete carbonization of organic precursors[11,22].

Furthermore, in the field of "traditional" organic synthesis, numerous studies have proved the crucial role of solvent parameters (hereafter referred to as "solvent effect"), including the dielectric constant, polarity, and basicity. Taking one step further, studies on the perspective of reaction kinetics and selectivity have revealed that the solvent influences on the reaction velocity, which may be described as a catalytic effect, and govern the transformation of various functional groups in organic molecules[23,24]. Unfortunately, despite its criticality, there have been only a few studies connecting this effect to the chemical modification of GQDs, which may reveal the mechanism of the doping profile changes in response to the solvent-assisted pyrolysis of organic precursors, e.g., glucose and citric/glutamic acids.

In this work, we report a single molecular precursor-based synthetic platform to synthesize chemically tailored N-doped graphene quantum dots (NGs) with high crystallinity and homogeneity that are self-passivated by various functional moieties with different chain lengths. Such an approach could provide a predictable synthetic tool for solvent-catalysts-aided growth of

heteroatom-doped GQDs via selective and quantitative C-N bond activation. By combining experimental results with quantum chemical calculations, we determine the reaction mechanism of oxidative activation of the nitrile groups in fumaronitrile (FN) in a variety of polar solvents, as well as the relationship between the structural features and optical characteristics of NGs using both steady-state and time-dependent density functional theory (SS- and TD-DFT, respectively), which can reveal the effects of end moieties as well as π-linkers. Finally, we conclude our study by exploring one of its practical applications, namely, light-sensitized photocatalysts. Specifically, to investigate the structural effects of NGs, we first apply GQDs for photocatalytic wastewater treatment, and more importantly, their potential medical application in ultralow-level energy photodynamic therapy (UL-PDT) is also validated. Consequently, the observed differences in the photocatalytic activities of each NG prepared from different solvent systems are consistent with our pre-estimations, intuitively reconfirming the effectiveness of our fabrication approach and the significance of theoretical prediction from both chemical and optical perspectives.

## Results and Discussion

**Synthesis and chemical characterization of NGs.** In our study, NGs were synthesized by a one-step self-assembly growth method in which the organic precursor FN was added to each of the following four different solvent media: DI water (DW), THF, butanal (Bu), and benzaldehyde (Be). As graphically summarized in Fig. 1, the complete self-assembly reaction of FN proceeds via two distinctive but simultaneous processes. Most of the primary reactant in FN hydrolysis, i.e., $H_2O$, is assumed to be supplied from moisture in the air within the reaction chamber, which is actively diffused or permeated into the reaction volume by increased pressure at high reaction temperatures. More detailed descriptions are provided in the Supplementary Information (see Supplementary Figs. 1–5 and Supplementary Discussion 1).

First, to confirm the variations in chemical composition and bonding of each NG under different solvent effects, X-ray photoelectron spectroscopic (XPS) and Fourier transform-infrared spectroscopic (FT-IR) measurements were performed (Fig. 2a–c; Supplementary Figs. 6–9). According to the XPS results in Fig. 2a, the change in the O/C atomic ratio of NGs follows a trend opposite to that for the N/C value (top of Fig. 2c). In addition, in terms of the chemical bonding ratios of the NGs, there is an inverse correlation between the concentration of oxidized C and that of nitrogenated C (see Fig. 2b and bottom panel in 2c), which was also confirmed by the FT-IR data (Supplementary Fig. 9). The chemical structures of two representative NGs (NG_DW and NG_Be: most and least oxidized forms, respectively) are depicted in Fig. 2e. Because of the differences in functional moieties of each QD, their pH values were found to be slightly different, as shown in Supplementary Fig. 10. These analyses show the feasibility of effectively modulating the chemical bonding configuration and composition of NGs during the thermolytic self-assembly reaction of FN by applying different solvent conditions.

**Theoretical study on the chemical activation of FN.** To theoretically support these experimental observations of variations in the chemical composition of NGs, we performed DFT calculations by using the Gaussian 16 program package (Fig. 2d and f). Complete chemical conversion of FN proceeds in two consecutive steps (Step 1: $C \equiv N \rightarrow CONH_2$ and Step 2: $CONH_2 \rightarrow COOH$), as depicted in Supplementary Fig. 11. First, the two most plausible mechanisms for the C–N activation of FN (FN → FAm (fumaramide-like molecule)) are depicted at the top of Fig. 2d,

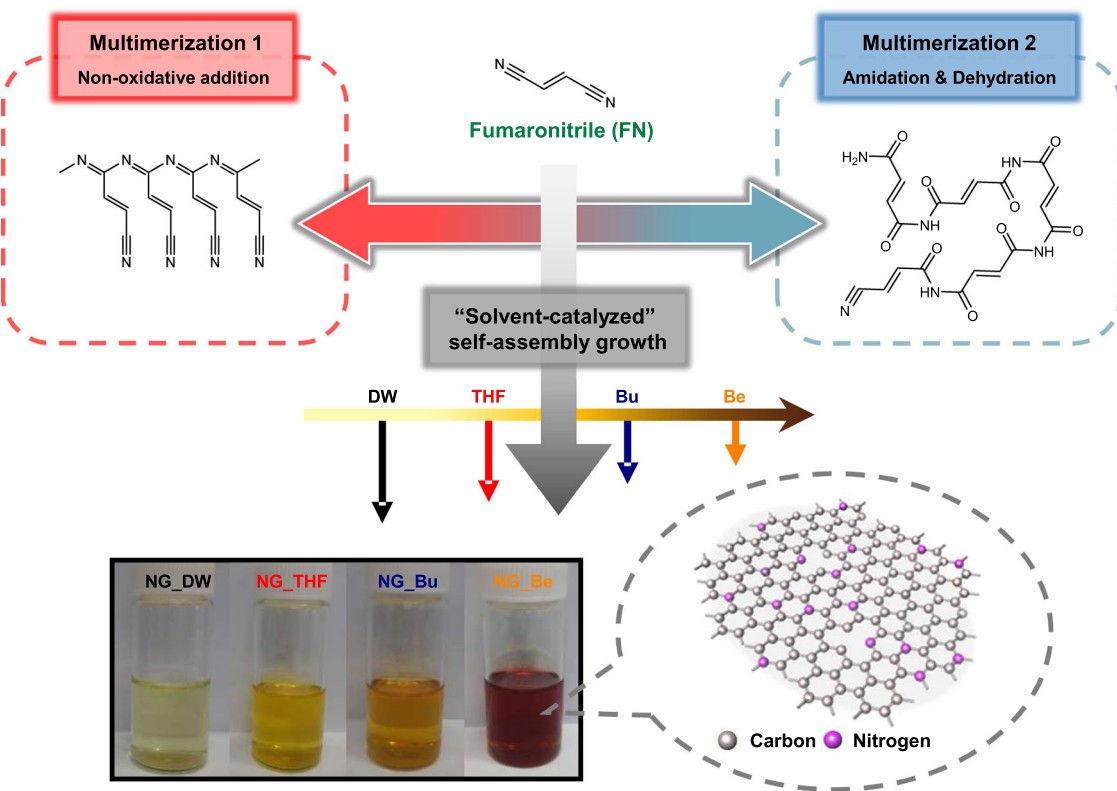

**Fig. 1 Illustration of the formation mechanism of chemically-tailored NGs.** Schematic diagram of the possible formation mechanisms of NGs through the thermolytic self-assembly reaction of FN, and photographs of their ethanol solution preparation under four different solvent systems.

where Paths I and II represent the general solvolysis reaction and solvent-catalysed oxidation reaction, respectively. Both transformation mechanisms of the C≡N functional group are fundamentally based on the addition-elimination mechanism, adding nucleophile molecules to electron-deficient C atoms.

From the thermodynamic numerical analyses in the Fig. 2d, the supporting solvent molecules (SSMs) in our study, including THF, Bu, and Be, were eliminated from the candidates of nucleophiles due to their low affinity to the electron-deficient C atoms. Furthermore, based on the free energy calculations in Supplementary Fig. 12a, oxidative activation of FN would rarely be achieved in circumstances where the SSMs intervened in the net reaction and the barrier energy was accordingly increased. Therefore, those SSMs were ruled out from the potential causes of the high crystallinity in our NGs, represented with a transmission electron microscopy (TEM) image in Fig. 3b.

Alternatively, as shown in Supplementary Fig. 12b, all SSMs more likely serve themselves as pseudo-electron-withdrawing moieties for the C≡N group in the reaction system ($\Delta H < 0$) and generate a cation state (C≡N$^+$). In other words, this result suggests that the solvent used is not only the medium where chemical reactions are performed, but it also actively involves in the chemical process. Furthermore, the change in Gibbs free energy ($\Delta G$) for this step is positive due to the negative contribution of the entropy change caused by the formation of H-bonds and increased van der Waals forces.

More importantly, the overall profiles of the potential energy surface of FN activation depend significantly on the types of solvents. In other words, as shown in the left of Fig. 2f, the change in $\Delta G$ is quite different in every elementary reaction for the transformation of the −C≡N group to its oxidized form. In particular, it is clearly observed that there is a distinct difference in the $\Delta G$ value of the rate-determining region (intermediate state structure #2 (IM2) — transition state structure #1 (TR1) & IM6

— TR2) in each reaction step. From the theoretical estimation of 60% higher activation barrier for TR1_Be than for TR1_DW, the corresponding pathway is considered relatively improbable and is simultaneously suppressed, regardless of its thermodynamic energy. The above thermodynamic and kinetic calculations signify that the SSMs can effectively control the oxidation degree of FN. Consequently, we can argue that the solvent in the reaction system does not only serve as a dispersing agent, but also it can be promoted as an active interventor which can alter the energy characteristics of the system during the intermediate and transition states of the synthesis.

Interestingly, it was found that the values of effective Gibbs free energy ($\Delta G_{\text{eff}}$, integration of $\Delta G$ provided in Fig. 2d, f) is observed to be a monotonic increasing function of the atomic ratio of resulting NGs (Fig. 2a), and thus this trend can be fitted by a bounded exponential growth curve as plotted in Fig. 2g. Reminding the importance of C≡N bond activation in either FN or its multimeric form for the formation of NGs, this correlation seems to be reasonable because their composition modulation can be attributed to the oxidation degree of FN. Also, we investigated the reliability and predictability of the model for the chemical composition of NGs prepared under other solvent systems (Supplementary Fig. 13), proving the universality and effectiveness of our approach for the production of chemically tailored GQDs. Therefore, based on the results obtained from analytical experiments and DFT calculations in this study, we are able to assert that our approach can be used as a predictable synthetic tool for application-specific NGs.

**Surface and structural analysis of NGs.** For further experimental verifications of the resulting NGs under different solvent effects, the surface morphologies of each sample were investigated by atomic force microscopy (AFM) and TEM. As seen from the

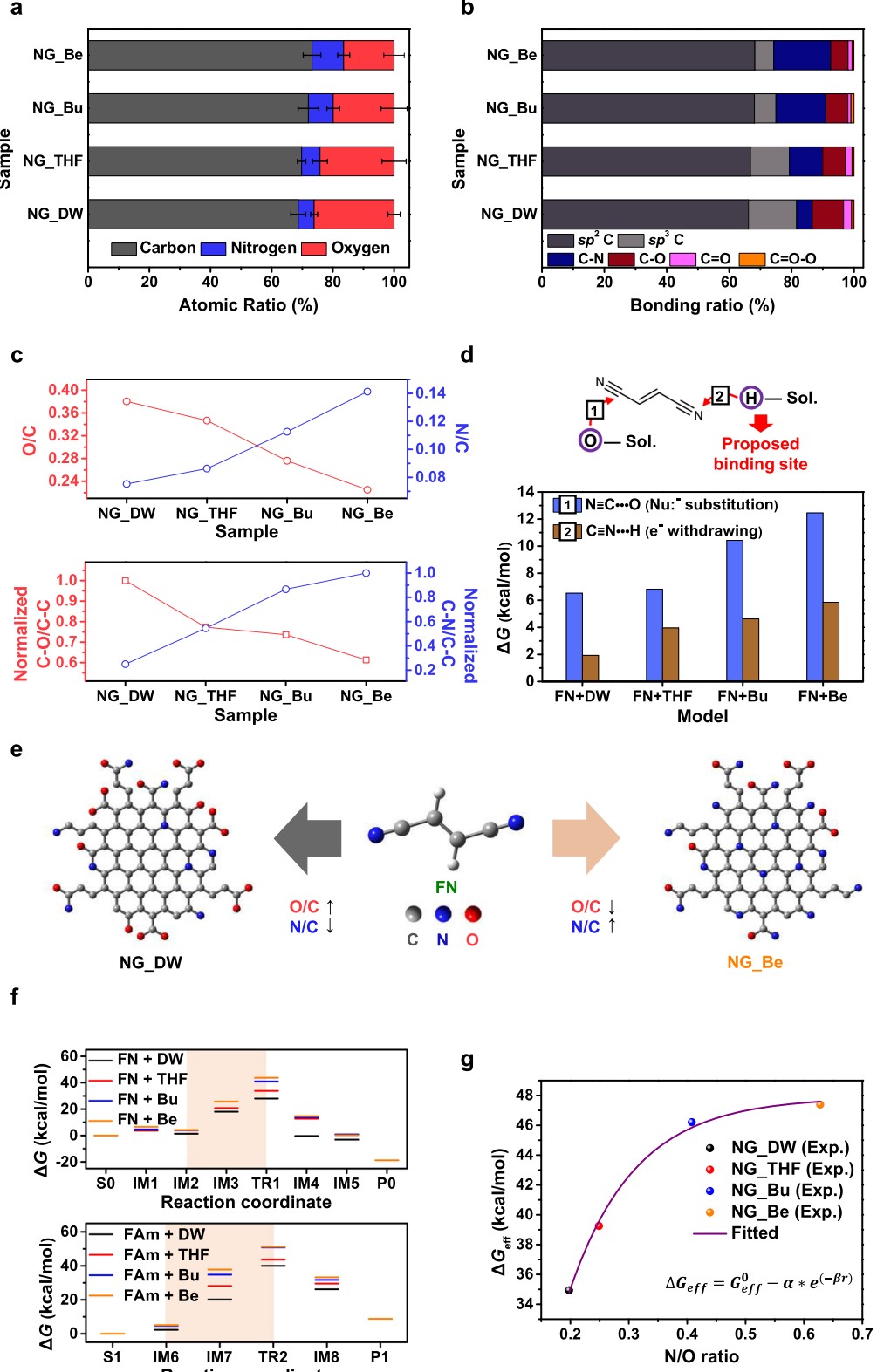

**Fig. 2 Chemical composition characterization of NGs. a** Elemental analysis from XPS measurements of NGs prepared with different reaction solvents. Error bars represent standard deviations from five independent experiments. **b** Multiple C-bonding configurations observed in the C 1$s$ spectra of each NGs. **c** Relative atomic ratios (O/C and N/C, top) and chemical bonding ratios of NGs (C-N/C-C & C-O/C-C, bottom). **d** Computationally derived interaction energies between FN and organic solvent molecules. **e** Plausible chemical structures of NGs prepared from different solvent systems. **f** Free energy profiles for the oxidative activation of FN (top) and FAm (bottom) calculated at the B3LYP/6-311 + G(d) level. **g** Correlation between the effective Gibbs free energy ($\Delta G_{eff}$), obtained from a weighted average of $\Delta G$ for selected steps, and the atomic ratio of resulting NGs ($\alpha = 75.6468$ kcal/mol, $\beta = 8.8652$, $G_{eff}^0 = 47.8776$ kcal/mol, $r = $ N/O ratio).

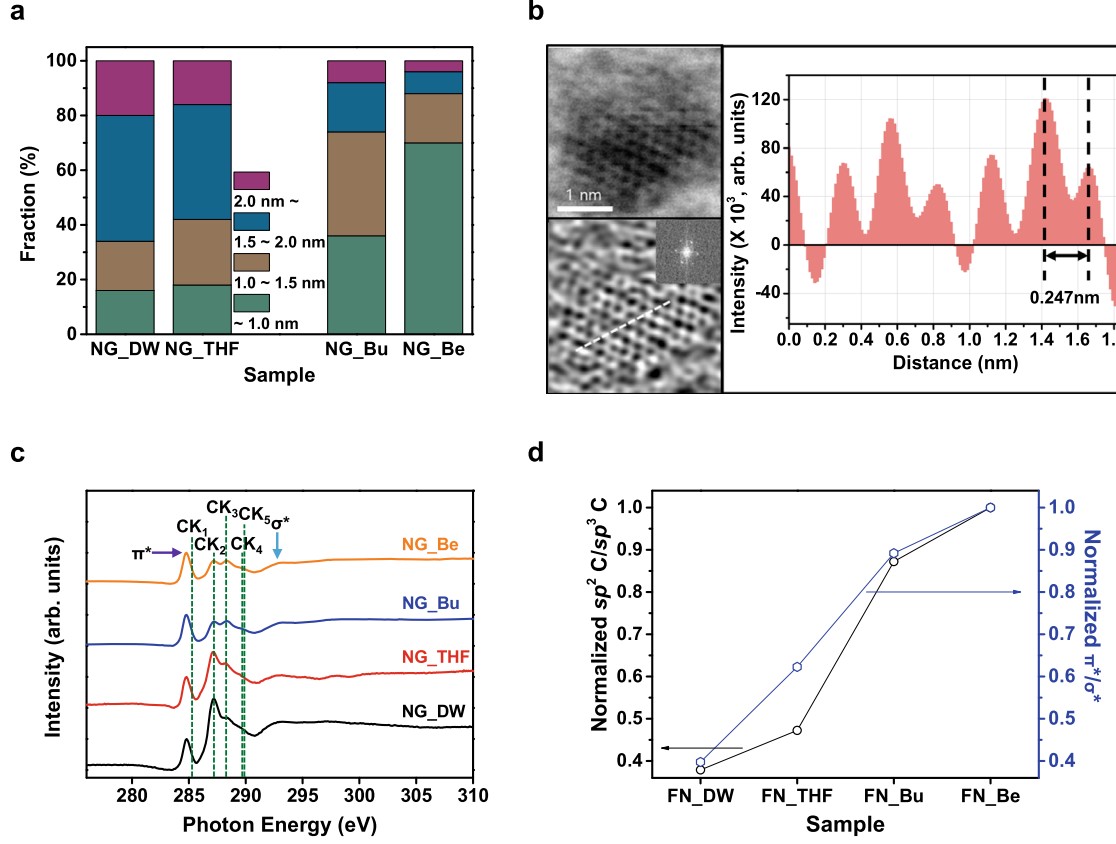

**Fig. 3 Morphological and structural characterizations of NGs. a** AFM-measured height distributions of NGs (collected at 50 dots per each sample). **b** Representative STEM image (top-left), corresponding IFFT image of a selected area (bottom-left, inset: FFT image), and intensity line profile (right) along the dashed line of NG_Be (atom distance of ~0.24 nm, matching with average distances in graphitic materials (11̄20))[25]. **c** NEXAFS spectra of four different types of NGs. The lowest energy resonance, at ~285.2 eV, corresponds to the transition of C 1s core-level electrons into states of $\pi^*$ symmetry around the M and L points of the graphene Brillouin zone above the Fermi level, whereas the higher energy peak at ~293.2 eV can be attributed to the excitation of C 1s core electrons into dispersionless states of $\sigma^*$ symmetry[46,47]. A more detail description is presented in the Method section. **d** Ratios of $sp^2$ C–C/$sp^3$ C–C (black circle) and $I_{\sigma^*}/I_{\pi^*}$ (blue hexagon) in NGs prepared in different reaction solvents.

AFM images (Supplementary Fig. 20), the heights of all samples range from 0.5 to 2 nm, suggesting the successful synthesis of NGs in all four different reaction conditions. Notably, the average height of NG_Be is found to be <1.0 nm (1–2 layers) in Fig. 3a, suggesting effectively suppressed vertical growth compared to the others. At this stage, we suppose that these features are likely related to the competitive reaction pathways for non-oxidative addition multimerization and solvent-assisted oxidation of FN (Supplementary Fig. 22). More discussion regarding the MG growth mechanism is provided in the Supplementary Information (see "Supplementary Discussion 2"). More analyses of the exact origin of the morphological variation are ongoing and will be included in our future studies.

The majority of NG_Be forms a monolayered graphene sheet, and its hexagonal lattice structure was readily observed by means of atomic-resolution scanning transmission electron microscopy (STEM), as displayed at the top left of Fig. 3b and Supplementary Fig. 21. From these data, it is clear that the formed zero-dimensional nanostructures are almost defect-free and highly crystalline, with diameters of <5 nm. These findings are also confirmed by the results of XPS and near-edge X-ray absorption fine structure (NEXAFS) analysis shown in Fig. 3c and d, respectively. In particular, the relative intensities of $\pi^*$ resonances ($\pi^*/\sigma^*$) in Fig. 3d provide an accurate estimate of the fraction of $sp^2$ (graphitic region) to $sp^3$ (amorphous defect sites) domain configurations, consistent with the XPS results[25]. Additional descriptions of the detailed investigation of the graphitization

mechanism are given in the Supporting Information (see "Supplementary Discussion 2").

**Experimental and theoretical studies on the optical properties of NGs.** To understand the changes in the optical properties of NGs as a result of the solvent effects, UV-visible absorption spectra were obtained, as shown in Fig. 4a (top) and Supplementary Fig. 23a. In addition to the two excitonic absorption bands (described in the caption of Fig. 4a), broad absorption peaks in the relatively lower energy band (350–500 nm) are also observed. These features are reflected in their photoluminescence excitation (PLE) spectra (Supplementary Fig. 23b) as well and consist of two electronic transitions at 260–270 nm ($\pi \rightarrow \pi^*$ excitation) and 360–380 nm (n $\rightarrow \pi^*$ excitation)[8].

In a theoretical determination of the correlation between the molecular structure and optical properties of the NGs, both SS- and TD-DFT calculations were performed by following the detailed procedure described in the Method section. The calculated absorption spectra of pristine and modified $C_7$-GQDs in ethanol (solvation model: SMD) are presented in Supplementary Fig. 14. Additionally, the calculated parameters of absorption energy, wavelength, and oscillator strength ($f$) for the first 5 lowest excited states of $C_7$-GQDs are listed in Supplementary Table 1. First, regardless of the types of heteroatom-containing substituents, all $C_7$-GQDs under consideration exhibited a redshift in their absorption wavelengths and much broader FWHMs than the pristine sample, as shown in Supplementary

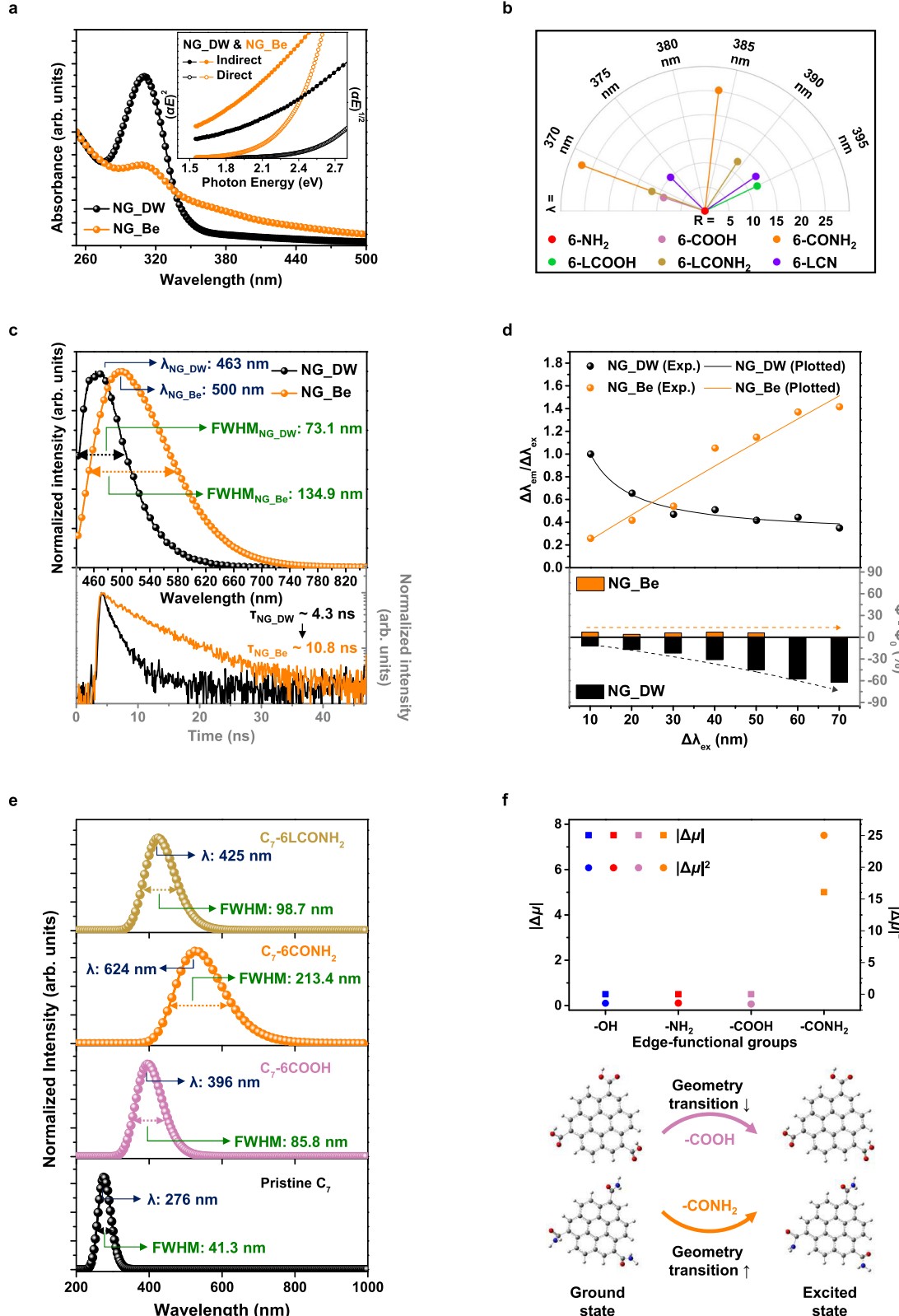

Fig. 14. These shifts can be ascribed to changes in the *f* values of the electronic transitions caused by the distorted molecular symmetry as well as the different electron distribution in the graphene moieties[26].

Specifically, depending on the oscillator strengths of $S_0 \rightarrow S_n$ for $C_7$-GQDs in Fig. 4b, the enhanced absorption capacity at longer wavelengths (350–500 nm) is significantly affected by the N-related surface states. In particular, according to the frontier molecular orbital (MO) and natural transition orbital (NTO) calculations in Supplementary Figs. 15–18, the -CONH$_2$ functional group at the edge of the NGs can induce an obvious decrease in the electron density delocalization in the HOMO and LUMO as well as spatial HOMO-LUMO separation, resulting from its highly distorted molecular structure[26]. Additionally, the

**Fig. 4 Experimental and theoretical investigations on the optical properties of NGs. a** UV-vis absorption spectra of NG_DW and NG_Be (top). Two excitonic absorption peaks: at <270 nm by the $\pi \rightarrow \pi^\star$ transition of aromatic $sp^2$ domains, at 310 nm by the $n \rightarrow \pi^\star$ transition of O- and N-containing bonds[48]. (inset: plots of $(\alpha E)^2$ and $(\alpha E)^{1/2}$ against photon energy ($E$) for the two types of NG solutions ($\alpha$: absorbance). **b** Calculated $f$ of $C_7$-GQDs with different types of edge-functional groups. The resulted magnitude of $f$-values is represented by the radius ($R = 10 \log(f/f_{6-NH2})$) and absorption wavelength is represented by the angular variation. **c** PL (top) and TRPL (bottom) spectra of NGs. **d** Comparison of emission peak positions (top) and QYs (bottom) of NGs as a function of $\lambda_{ex}$. **e** Theoretically simulated PL spectra of $C_7$-GQDs with different types of edge-functional groups. **f** Theoretical calculation of ground and excited-state dipole moments of the $C_7$-GQDs with different edge-functional groups (top). Frontier molecular orbitals for $C_7$–6COOH and $C_7$–6CONH$_2$ at the optimized geometries (bottom).

existence of a self-passivated $\pi$-linker, namely, a vinyl bridge, can provide a much larger $\pi$-conjugation domain to ensure effective $\pi$ delocalization with strong extended intramolecular charge transfer[27]. From these observations, the disruption of the structural symmetry or elongation of the $\pi$-conjugation was revealed to cause splitting in the $\pi$ and $\pi^*$ levels with a decrease in the energy gap between the HOMO and LUMO, as shown in Supplementary Fig. 19, leading to a broadening of the absorption band towards the longer wavelength region[28].

The corresponding PL spectra of each NG are presented in Supplementary Fig. 24, exhibiting a strong dependence on the types of edge-functional moieties. In detail, as shown in Fig. 4c, d, and Supplementary Figs. 24–26, there are different PL characteristics in terms of peak wavelength, FWHM, quantum yield (see "Supplementary Discussion 3"), and lifetime depending on the type of NGs. Based on the abovementioned chemical composition and structural analyses, these observations are assumed to be an outcome of the combined and interactive effects of (1) internal charge transitions and (2) solvation dynamics, which are attributed to the changes in the edge or surface states induced by the heteroatom-doping profile.

First, the overall photoluminescence (PL) features of each NG in Fig. 4c and d are ascribed to a disruption of the structural symmetry or elongation of the $\pi$-conjugation caused by the presence of functional groups on the material surfaces. On the basis of our calculations, the introduction of edge-functional moieties in NGs can induce substantial redshifts of their emission characteristics, as shown in Fig. 4e, which are associated with the modifications of their inter- and intra-band electron transition processes[29–31].

Another cause of the difference in $\lambda_{ex}$-dependent PL characteristics seen at top of Fig. 4d originates from the correlation between molecular configuration (chemical and geometrical features) and the dipole moment of the molecules[32,33]. According to the Lippert equation (see "Supplementary Note 1")[32], such phenomena can be explained by the solvent relaxation and dipolar reorganization effect of the vicinal solvent molecules, where the solvent effect governs changes in the dipole moment (fluorophore) under PL excitation[34].

As mentioned above, the presence of –CONH$_2$ functional groups at the surface of NGs can lead to strong intramolecular charge migration, resulting in a maximum change in the dipole moment between the ground and excited states among various chemical configurations (Fig. 4f). Note that, for simplicity, the chain length effect was omitted from the estimation of the dipole moment induced by edge-functional groups. At this point, to reduce the interaction between the excited state dipole and the solvent molecules induced by this transition, the solvent cage surrounding the QD surface is inevitably relaxed, which causes both inhomogeneous broadening and a redshift of the PL spectra.

**Photocatalytic degradation test and analysis.** In conjunction with all of the above confirmation of consistency between our theoretical predictions and the experimental results, the effectiveness was verified to strengthen the feasibility of our work in that, by manipulation of the QD's inherent characteristics, their

practical uses can be expanded. In terms of our proposed platform for diverse opto-chemical applications, we monitored NG-assisted photocatalytic degradation of rhodamine B (RhB) and uric acid (UA), where photodecomposition in the latter case can potentially be used for the in vivo cure of metabolic diseases, e.g., Lesch-Nyhan syndrome, gout, and cardiovascular disease[35,36].

For the photocatalytic test, upon exposure to a broadband visible light source (400 nm longpass filtered xenon lamp, 38.6 mW·cm$^{-2}$), temporal degradation profiles of both target molecules (in aqueous media), which were physically adsorbed onto the surface of either the NGs (shell) or the TiO$_2$ NPs (core), were measured. As a pre-estimation of the above catalytic effects, possible initial degradation sites in each pollutant (marked with red circles in Fig. 5a) were theoretically elucidated by calculations involving Fukui functions ($f^0$) and accordingly resulted in values of local softness ($s^0$) for radical attack[37,38], where higher photodegradation efficiency for RhB is expected from its higher $s^0$ value. Additionally, the number of radicals generated during photocatalysis strongly reflects the light absorption capacity of the catalysts (more description in Supplementary Discussion 4), and among the different types of NGs in our study, NG_Be is expected to exhibit the highest catalytic activity for the degradation of organic pollutants (Fig. 4a).

In the experimental results shown in Fig. 5c, d, similar trends of photocatalytic effects were observed, where the pseudo-first-order rate constants ($k$) of the degradation curves with TiO$_2$@NG_Be were found to be 0.269 and 0.193 hr$^{-1}$ for RhB and UA, respectively. In contrast, when NG_DW was used as a shell material, as predicted by the above theoretical analyses, the decay constants of degradation decreased slightly to 0.0929 and 0.0548 hr$^{-1}$ due to the lower light-harvesting capacitance of this material. Remarkably, when pure TiO$_2$ NPs were used for the removal of RhB, the decay constant was as low as $k = 0.0179$ hr$^{-1}$ (Supplementary Fig. 27) due to the wide intrinsic band gap[39], but, on the other hand, the core-shell NPs fabricated by introducing only 0.5 wt% NGs onto the TiO$_2$ surface could substantially enhance the rate of photocatalytic activity up to ~16 times. Considering that the influence of NGs incorporated into the surface of TiO$_2$ on its morphology and phase structure is negligible, the enhanced photocatalytic activity can be attributed to the photo- and electrochemical properties of the materials.

With respect to charge kinetics, the mechanism of the above enhancement can also be elucidated by the synergistic effect of quaternary N (N$_q$) and pyridinic N (N$_{pydi}$), as shown in Fig. 5b. Specifically, N$_q$ can serve as an electron transfer facilitator for the fast and effective transfer of photogenerated electrons to the photodegradation active site because it aids in supporting electron delocalization effects without destroying $sp^2$-hybridized graphitic structures[40]. On the other hand, N$_{pydi}$ at the edges or defects can play a dual functional role as a strong electron-withdrawing group, which results in a more negative electrostatic potential (ESP) charge, as well as a catalytic active site for the removal of organic pollutants[41]. Furthermore, the C atoms in the vicinity of N$_{pydi}$ can easily capture H$_2$O and O$_2$ molecules with high $s^+$ and subsequently form the catalytic molecules •OH$^-$ and •O$_2^-$.

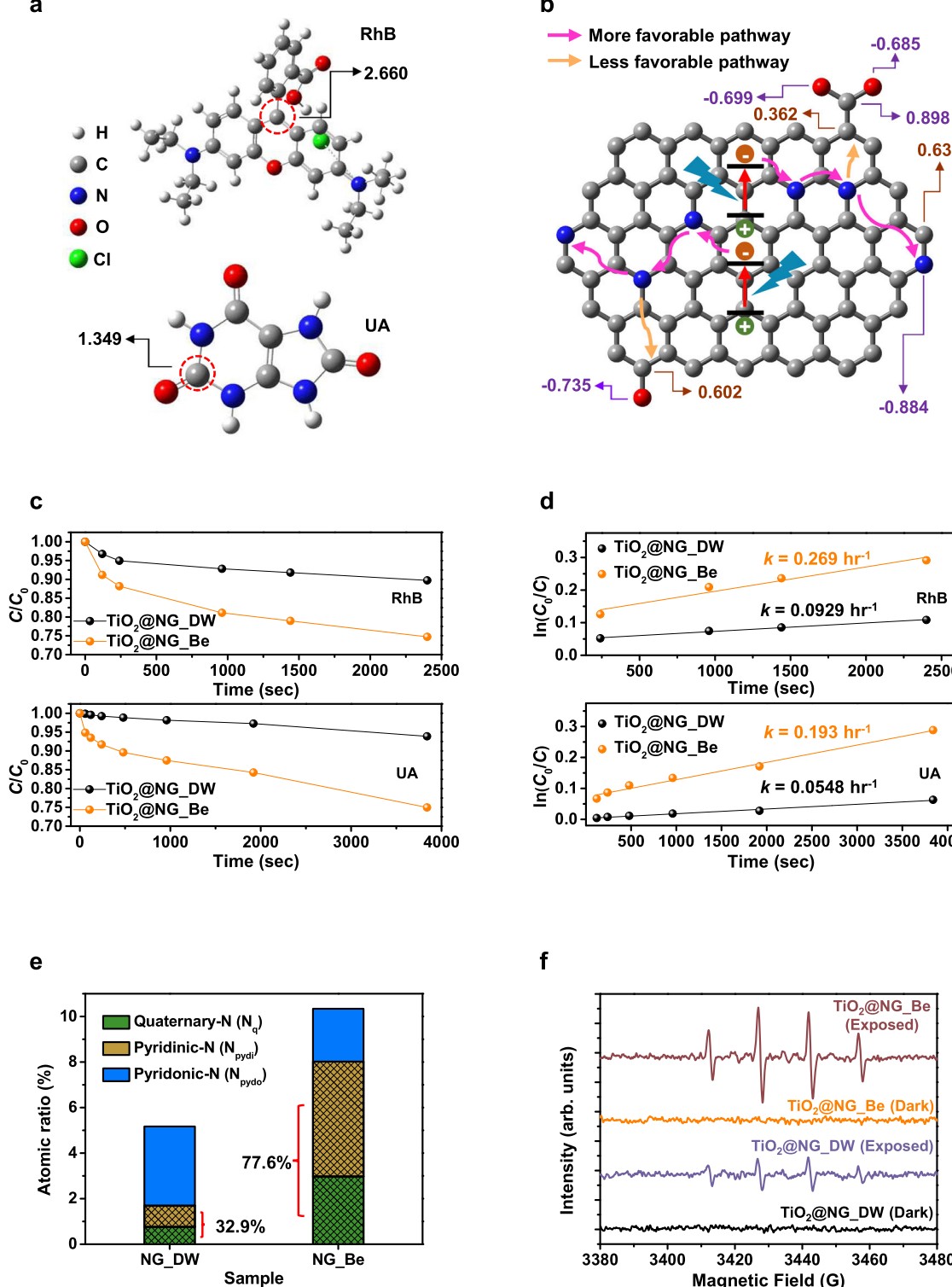

**Fig. 5 Photocatalytic activity of NG-decorated TiO₂ composites. a** Optimized chemical structures of RhB (top) and UA (bottom). Their local softness values ($s^0$) were calculated using the NBO analysis. **b** Calculated local softness values ($s^+$ in brown) of the N-doped graphitic carbon nanostructure. Atomic charges in purple were obtained from the ESP fitting according to the Merz-Kollman approach at the B3LYP/6-311 + G(d) level. **c** Photocatalytic degradation of RhB (top) and UA (bottom) in the presence of TiO₂@NG_DW and TiO₂@NG_Be catalysts under visible light irradiation (λ > 400 nm, 38.6 mW cm⁻²). **d** First-order kinetics of RhB (top) and UA (bottom) photocatalytic degradation by different photocatalysts. **e** Chemical compositions of N-containing groups in two types of NGs, obtained from N 1$s$ XPS spectra. **f** EPR spectra of photocatalyst in aqueous dispersion for DMPO-•OH⁻ under dark and visible light irradiation conditions.

From another perspective, substituting N atoms in the basal plane of NGs is known to induce the formation of n-type domains, which are expected to facilitate the free charge carrier distribution of NGs, which is attributed to the coexistence of p- and n-type domains in the QDs[42]. Additionally, due to their nanoscale size, this configuration substantially leads to changes in the near-surface electric field, which has a significant influence on intramolecular photoinduced charge separation (Supplementary Note 2). Accordingly, p- and n-type regions in graphitic structures can be regarded as an important factor in improving exciton separation and carrier migration efficiencies in NG-assisted photocatalysis systems. As shown in Fig. 5e and Supplementary Fig. 8, each NG has a different N content and N-bonding configuration. NG_Be contains the largest amount of both $N_q$ and $N_{pydi}$, as well as the highest N/C (0.14) and N/O (0.63) atomic ratios, implying that TiO$_2$@NG_Be may show not only the highest internal charge dissociation efficiency of photogenerated hole-electron pairs but also the highest generation rate of the free radicals.

To verify these hypotheses, an electron paramagnetic resonance (EPR) spin-trap measurement was performed by using DMPO (5,5-dimethyl-1-pyrroline-N-oxide) as a spin-trapping agent. As shown in Fig. 5f and Supplementary Fig. 28, no DMPO-•OH spin adducts were observed in the dark for either TiO$_2$@NG_DW or TiO$_2$@NG_Be, whereas the characteristic signals for DMPO-•OH (1:2:2:1 quartet pattern) emerged under visible light irradiation. More importantly, it can be clearly seen that •OH signal of TiO$_2$@NG_Be is much stronger than that of TiO$_2$@NG_DW. From these observations, we can conclude that the chemical configurations of NGs could modulate the generation rate of active radicals for the photocatalytic degradation of organic pollutants.

In summary, we developed a "creative" approach to synthesize customizable NGs with a highly ordered graphitic structure via the one-step thermolytic self-assembly reaction. Oxidative activation of the nitrile groups in FN is effectively controlled by the type of solvent, leading to changes in the elemental compositions (N/C and N/O ratios) and N-bonding configurations ($N_q$, $N_{pydi}$, $N_{am}$ and $N_{pydo}$) of NGs. The differences in the chemical configurations of NGs strongly influenced their optoelectronic behaviours as well as their surface morphology features.

Using various computational methods, in regards to the kinetics and thermodynamics of FN activation, we systemically confirmed the solvent catalytic effects on the formation of NGs. As another verification of our reaction mechanism, we also investigated the correlation between the structural features and optical characteristics of NGs.

Finally, it was clearly observed that the introduction of NGs onto the surface of TiO$_2$ has little influence on its structure and morphology, but it significantly enhances the efficiency of light harvesting, as well as charge separation of the photocatalyst due to their N-related chemical components. From these findings, our future studies are expected to provide a variety of innovations in the field of opto-electronics, biological applications, and chemical sensors.

## Methods

**Preparation of NGs**. NGs were synthesized by thermal treatment of FN powder purchased from the Tokyo Chemical Industry. 0.1 g of FN was dispersed in 10 mL of each solvent (DI-water, tetrahydrofuran, butyraldehyde, and benzaldehyde), injected into a 100 mL Teflon liner, and then heated to 170 °C for 10 min in a stainless-steel reactor using a hot plate while stirring continually to obtain stable and uniformed products. The colours of the liquid consequently changed from colourless to pale yellow (NG_DW), normal yellow (NG_THF), dark yellow (NG_Bu), and dark red (NG_Be), indicating the successful formation of NGs. After the excess solvent was removed by vacuum evaporator (or vacuum oven), the final products (NGs) were recovered by freeze-drying process.

**Instrumentation and measurements**. AFM imaging was performed in the air using an NX-10 instrument (Park systems). Cs-corrected STEM (Cs-corrected STEM) images of NGs were obtained with a JEM-ARM200F operated at an accelerating voltage of 80 kV and equipped with a cold field emission gun (CFEG). UV-vis absorption and PL spectra were acquired by a Jasco V-670 spectrophotometer and a Horiba NanoLog-C, respectively. FT-IR spectra were obtained by an FT-IR spectrophotometer (Jasco FT/IR-6600). [1]H NMR spectra were recorded in CDCl$_3$ at 600 MHz (Agilent 600 MHz Premium COMPACT NMR Magnet). Raman spectra were obtained using a Horiba Jobin Yvon-Labram HR UV-Visible-NIR Raman microscope spectrometer. NEXAFS measurements were conducted at the soft X-ray 10D XAS KIST (Korea Institute of Science and Technology) beamline. Each spectrum of NGs was obtained in total electron yield mode at room temperature and ~ 10$^{-8}$ Torr, and they were then normalized with respect to the incident photon flux. All spectra were acquired at an incidence angle of 54.7°, where the peak intensities are independent of the angular symmetry dependence. The $\pi^*$ resonance is representative of the unsaturated C=C or graphitic ($sp^2$) bond above and below the plane, including the C–N transitions (CK$_1$, 285.4 eV), while the in-plane $\sigma^*$ feature is typical of a C–C bond[43,44]. The four resonances in the intermediate region between the $\pi^*$ and $\sigma^*$ features were observed at ~287.2 eV (CK$_2$), ~288.3 eV (CK$_3$), ~289.1 eV (CK$_4$), and ~289.5 eV (CK$_5$) and were assigned to functional groups (CK$_2$: C–O, CK$_3$: C=O, CK$_4$: O–C=O and CK$_5$: various amide groups) that decorate the basal planes and the edge sites of NGs[45]. XPS analysis was carried out using a monochromated Al K$\alpha$ X-ray source ($h\nu = 1486.6$ eV). To measure the electron lifetime of NGs, TRPL spectroscopy was employed. The second harmonic generation (SHG = 370 nm) of a tunable Ti:sapphire laser (MaiTai, Spectra-Physics) with an ~100 fs pulse width and 80 MHz repetition rate was used as the excitation source. EPR measurements were performed using a CW-EPR (microwave frequency and power of 9.64 GHz and 3 mW, respectively, and modulation frequency of 100 kHz).

**Preparation of TiO$_2$@NG powders**. The powders were prepared by a mild temperature hydrothermal process to avoid further oxidation of NGs. Typically, TiO$_2$ (Aeroxide® P25, 20 mg) was added to 10 mL of NG aqueous solution (0.01 mg/mL), which was then incubated for 24 hr at 70 °C to decorate NGs on the TiO$_2$ surface (agitation speed: 400 rpm); excess DI-water was removed by utilizing a vacuum evaporator. The final products (TiO$_2$@NGs), which contain 0.5 wt% of NGs, were collected after the drying process.

**Photocatalytic degradation test**. Photocatalytic activities of TiO$_2$@NG powders were evaluated by monitoring the degradation of RhB and UA under the illumination of 38.6 mW/cm$^2$ of white light (<400 nm cut-off). The photocatalytic experiments were carried out at room temperature. In order to investigate the photodegradation efficiency of the catalyst, the aqueous solutions before/after the photodegradation reaction were analysed by absorption and fluorescence techniques as a function of reaction time.

**TD-DFT calculations**. The vertical excitations of the lowest five excited states were calculated using TD-DFT with CAM-B3LYP/6-31 + G(d) basis set. To clearly describe the diffuse excited states, we chose a double zeta basis set including polarization and diffuse functions, 6-31 + G(d). The ground state geometries of the models were optimized at B3LYP (or CAM-B3LYP)/6-31 + G(d) level of theory and confirmed by frequency calculations (Opt + Freq). Also, the relaxation of the geometries in S1 was performed at TDDFT-CAM-B3LYP/6-31 + G(d) level of theory. The solvent (solvent: ethanol) is included based on the Polarizable Continuum Model (PCM). The optical bandgaps of each model were calculated to be the difference of energy between the ground and the first excited states. The atomic charges in the ground and the first excited states were obtained using the natural bond orbital (NBO) method. All calculations were carried out by Gaussian 16 software package. First, we modelled and simulated several local structures of NGs with a various number of aromatic rings (C$_x$-NGs, $x$ = 1, 2, 4, 7, 19, and 36), including different heteroatom functional groups at the edge sites. For the accurate but cost-efficient calculations, we finally selected pristine and edge-functionalized C$_7$-GQDs.

## Data availability

The authors declare that the findings that support the findings of this study are available within the article and Supplementary Information files, and the extra files are also available from the corresponding author (sbae@kist.re.kr, response timeframe: 1 week) upon reasonable request. The data are not publicly available due to privacy or ethical restrictions. Also, the raw/processed data required to reproduce these findings cannot be shared at this time as the data also form part of an ongoing study.

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

## Acknowledgements

This work was financially supported by the Korea Institute of Science and Technology (KIST) Institutional Program, the Ministry of Trade, Industry & Energy of Korea (20011317), and the National Research Council of Science & Technology (NST) grant (CRC-20-01-NFRI).

## Author contributions

B.J.M. and S.B. conceived the idea and designed the experiments. B.J.M., S.J.K., A.L. and Y.O. synthesized and characterized the samples. B.J.M. and B.H.H. performed the DFT calculations and analysis. B.J.M. and S.H.L. carried out TRPL measurements. S-K.L. and T-W.K. conducted the NEXAFS measurement. B.J.M. and Y.O. performed photocatalytic experiments. B.J.M., S.J.K., A.L. and S.B. wrote the manuscript and prepared the figures. All authors participated in discussions and commented on the manuscript.

## Competing interests

The authors declare no competing interests.
