## [Peer Review File · Nature Communications]

Structure-controllable growth of nitrogenated graphene quantum dots via solvent catalysis for selective C-N bond activationReviewers' comments:

Reviewer #1 (Remarks to the Author):

This paper presents a creative approach to produce structure-controllable NGs via one-step solvent-catalysts-aided thermolytic self-assembly of a single molecular precursor. The solvent catalytic effects on the formation of NGs, and the correlation between the structural features and optical characteristics of NGs, was evidenced by theoretical calculations. However, I think that it cannot be suitable for publication in Nature Communications. There are some questions as follows:

1. The authors asserted that this approach can be used as a predictable synthetic tool for application-specific NGs, however, there is a lack of evidence. It should better be testified by more theoretical prediction and experimental data with other solvents.
2. The results demonstrated that the composition of NGs could be modulated by the oxidation degree of FN, which was in turn decided by the solvents via solvent-catalysed oxidation reaction. Since H₂O is the leading reactant of FN hydrolysis, I think the water content, as well as oxygen, in the reaction system might also influence the oxidation degree of FN, which has not been taken into account in this work.
3. I wonder if the reaction temperature and time could play a role in the structure and optical properties of the resultant NGs. Will the change in reaction parameters lead to deviation between theoretical estimation and experimental results?
4. This work showed a strong dependence between the optical properties and the types of edge-functional moieties. Previous studies also verified the effect of size distributions and N-bonding configurations on the optical properties of graphene quantum dots. These should also be considered in this work.

Reviewer #2 (Remarks to the Author):

The paper describes a solvent-mediated growth of graphene quantum dots using a single source precursor procedure integrating some experimental evidences with a stronger theoretical calculation. I

believe that the work will positively contribute to active and practical utilization of graphene based nanomaterials in several applications. I recommend accepting the work but I would suggest authors to make a major revision by the following comments and considerations.

1- The chemical approach described in this paper is not substantially different for the experimental procedure reported elsewhere [Byung Joon Moon.... and Sukang Bae, Chem Mater 2016, 28, 1481-1488] where the synthesis of nitrogenated graphitic carbon dots has been described (“.0.3g FN dispersed in 10 mL DI water was injected into a 50 mL round bottle flask, and then heated to 225°C for 10 min ...”...while in the current paper... “0.1 g of FN was dispersed in 10 mL of each solvent injected into a 100 mL Teflon liner, and then heated to 170°C for 10 min”..). Considering that the differences observed in the current/submitted paper are from authors ascribed to supporting solvent molecules (SSM effect) that act as pseudo-electron-withdrawing moieties for C≡N group and/while the primary precursor (FN) activation is assumed to be supplied from air moisture within the reaction chamber, how do they think that the described synthetic mechanism would be influenced by different reaction conditions such as an increasing the concentration of FN and/or an increasing the reaction temperature? In other words, which is the main reason that allows the production of graphene and graphite with the two different while similar synthetic procedures since the FN overall concentration is not extremely different within the two procedures? If they would consider the synthetic temperature as the main factor that would preferentially promote a bidimensional (1-2 layered NGs) or “three dimensional” (4 layered NGs) as described in Scheme S1, do they have any experimental result about the here reported nitrogenated graphene quantum dots synthesis in presence of benzaldehyde, but heating the system at 225°C for 10 minutes?.

2- In the abstract and into the introduction authors describe their procedure as scalable (line 27) and self-assembled reaction (line 92). I haven't find any concrete indication, evidence or purely explanation in the paper that the procedure is effectively scalable at larger quantity reaction batches. Moreover, the self-assembly regards a specific concept concerning the formation of highly ordered structures, in which, ideally, diffraction methods allow one to deduce structural periodicities to quantify the order beyond a coarse microscopic observation. My feeling the self-assembly of FN is not the correct way to describe the system and is not concretely supported in the paper, whereas the reaction is more a thermolytic polymerization rather than a chemical self-assembly.

3- The oxidative activation of C≡N group is activated by moisture. Is there any specific reason or explanation that allows the partial oxidation of only one C≡N group of FN molecule and not both C≡N groups?

4- In the main text of the paper, are often reported some abbreviations not previously described that makes the reading uncomfortable.

a. Line 148: Fam that I believe is fumaroaldehyde-like molecule, but it is not explicitly announced in the text.

b. Line 175 and Fig1f, acronyms as IMs, TR1, IM6, TR2, etc..that I believe to be reaction intermediates, but are not at all described or explained, I think it makes the paper not clearly understandable from a reader.

5- Few figures reported in the supporting material are not even mentioned in the main text, such as Fig S1 (Raman) and Fig S2 (NMR) which might contain in principle a more detailed information but are not used in the main text, making them useless. Fig S3, which is only noise or a not-emissive sample, should be integrated of numerical counts on y axis in the way of making clearer that represent a “zero” value.

Reply to Reviewer #1

This paper presents a creative approach to produce structure-controllable NGs via one-step solvent-catalysts-aided thermolytic self-assembly of a single molecular precursor. The solvent catalytic effects on the formation of NGs, and the correlation between the structural features and optical characteristics of NGs, was evidenced by theoretical calculations. However, I think that it cannot be suitable for publication in Nature Communications. There are some questions are as follows:

(Comment #1) The authors asserted that this approach can be used as a predictable synthetic tool for application-specific NGs, however, there is a lack of evidence. It should better be testified by more theoretical prediction and experimental data with other solvents.

Response:

Thanks for the reviewer's consideration. In this study, for the determination of solvent catalytic effect on the conversion of fumaronitrile (FN), as a validation, its underlying mechanism of intermolecular interaction between FN and solvent catalyst (SC) was sufficiently verified by chemical analysis, e.g., XPS, NMR, and FT-IR, as well as DFT calculations with B3LYP/6-311 + G (d) level. In this step, we preliminarily confirmed that, through the same underlying mechanism, chemically-tailored NGs are successfully prepared in a variety of solvent systems, including those not mentioned in this paper (Figure R1). Among them, representative four types of solvent systems were selected to identify the correlation between the activation free energy of -CN (& -CONH₂) and the atomic ratio of resulting NGs more thoroughly and to further investigate their optical and photochemical characteristics. Also, among various samples, we chose NGs with entirely different chemical compositions (e.g., N/O ratio) from each other, which was to ensure the high reliability of the proposed model.

Consequently, although the reviewer mentioned a lack of evidence on the generality, we believe that our proposed model was sufficiently supported by the data presented in the manuscript. Nevertheless, as requested by the reviewer, we acquired sufficient and explicit evidence (e.g., chemical composition, free energy profiles) to concretely prove the universality of our strategy for the synthesis of chemically-tailored GODs (Figure R2).

Figure R1. Photographs of various NGs dissolved in ethanol, which were (a) included or (b) excluded in manuscript. (NG_MF: *N,N*-Dimethylformamide, NG_EF: Ethyl formate, and NG_MP: *N*-Methyl-2-pyrrolidone)

Figure R2. (a) Applicability test of our proposed model (depicted in Figure 1g) for use as a predictable synthetic tool. (b) XPS survey spectra of NGs.

(Additional figure in supporting information)

Figure S11. (a) Applicability test of our proposed model (depicted in Figure 1g) for use as a predictable synthetic

tool. (b) XPS survey spectra of NGs (NG_MF: N,N-Dimethylformamide, NG_EF: Ethyl formate, and NG_MP: N-Methyl-2-pyrrolidone).

(Complemented manuscript)

[page 8, line 17] Reminding the importance of C≡N bond activation in either FN or its multimeric form for the formation of NGs, this correlation seems to be reasonable because their composition modulation can be attributed to the oxidation degree of FN. Also, we investigated the reliability and predictability of the model for chemical composition of NGs prepared under other solvent systems (Figure S13), proving the universality and effectiveness of our approach for the production of chemically tailored GQDs. Therefore, based on the results obtained from analytical experiments and DFT calculations in this study, we are able to assert that our approach can be used as a predictable synthetic tool for application-specific NGs.

(Comment #2) The results demonstrated that the composition of NGs could be modulated by the oxidation degree of FN, which was in turn decided by the solvents via solvent-catalysed oxidation reaction. Since H₂O is the leading reactant of FN hydrolysis, I think the water content, as well as oxygen, in the reaction system might also influence the oxidation degree of FN, which has not been taken into account in this work.

Response:

Thanks for your kind consideration. We also agree to a certain extent with the reviewer's comment. Regarding this comment by the reviewer, our perspective, as well as the previous considerations to verify the effect of these two factors, e.g., oxygen and moisture, are as below:

At first, although it was well-known that the C≡N group exhibited no or little reactivity with O₂ molecules at even high temperature (@ 500 °C).^[2-4] we already investigated experimentally that the effect of oxygen on the oxidation degree of C≡N group could be negligible at previous^[1] and present reaction conditions (170 °C, 10 min). Specifically, to prove this statement (result: Figure R3a), the GQD synthesis was performed using both i) reaction containers (e.g., Teflon liners) and ii) solvents in which the O₂ molecules were removed and replaced with N₂ gas (99.999% purity).

Second, considering the influence of water on the hydrolysis of nitrile groups in the reaction system, in this experiment, all solvents used were either purchased as anhydrous grade or were dehydrated using either sulfates or molecular sieves, which are commonly utilized for drying polar aprotic solvents. On top of that, for the verification of the effect of water content on the activation degree of nitrile groups in the reaction system, in this experiment, the synthesis of QGDs has already been carried out under excessive moisture condition (e.g., using water as a solvent, product: NG_DW). Also, as already described in the supplementary information (Section S2, page 4) “Even in a rather dry atmosphere, the shortage can still be supplemented by the H₂O molecules generated during the condensation polymerization or cyclization of C≡N groups in intermediate oligomers via the dehydration process.”, even in a rather dry atmosphere (~ 26%), under different solvent systems, chemically-tailored NGs were still obtained without any post-purification process, and their chemical characteristics were almost same for the samples with a relatively humid condition (~ 69%) in Figure R3b.

From this result, it was confirmed that, although the oxidation degree of FN is likely to be affected a little by the RH, it can be more greatly affected by the type of solvent system (Figure R3c). Also, taking one step further, we are able to assert that our strategy can be used as an efficient approach to systemically prepare chemically-tailored QGDs under ambient conditions.

Figure R3. (a) Chemical compositions of NG_THFs prepared under ambient (left) and N₂-degassed systems (right). (b) Chemical compositions of NG_THFs prepared under different relative humidity air conditions (26~69%). (c) Chemical compositions of NGs prepared under different solvent systems.

Figure R4. XPS survey spectra of NG_THFs synthesized under different reaction systems ((a) N₂-degassed system, (b) RH ~ 26%, and (c) RH ~ 69%).

(Complemented supporting information)

1) [page 3, section S2] Even in a rather dry atmosphere, the shortage can still be supplemented by the H₂O molecules generated during the condensation polymerization or cyclization of C≡N groups in intermediate oligomers via the dehydration process. In this regard, the effect of relative humidity (26% ~ 69% RH) of air on the activation degree of FN was investigated by adding water vapor to a fixed concentration of FN (Figure S4a). From this result, it was confirmed that, although the oxidation degree of FN can be slightly affected by the RH, it can be more greatly affected by the type of solvent system. Consequently, due to the reliable supply of H₂O molecules by these mechanisms, further supplement of water into the synthesis, e.g., use of water as solvent, is found to be unnecessary. Also, based on previous reports^{10,11} and our results (Figure S4b), the effect of oxygen on the oxidation degree of C≡N group would be negligible.

Figure S4. (a) Chemical compositions of NG_THFs prepared under ambient (left) and N₂-degassed systems (right). (b) Chemical compositions of NG_THFs prepared under different relative humidity air conditions (26~69%). (c) XPS survey spectra of NG_THFs synthesized under different reaction systems (top: N₂-degassed system, middle: RH ~ 26%, bottom: RH ~ 69%).

(**Comment #3**) I wonder if the reaction temperature and time could play a role in the structure and optical properties of the resultant NGs. Will the change in reaction parameters lead to deviation between theoretical estimation and experimental results?

(**Comment #3_1**) I wonder if the reaction temperature and time could play a role in the structure and optical properties of the resultant NGs.

Response:

Thanks for the reviewer's consideration.

First of all, you have raised an important point; however, we believe that the studies on the dependence of either reaction temperature or time on the optical or structural properties of NGs would be outside the scope of our study because, in this paper, we focused on the availability of this platform to be used as a predictable synthetic tool for the acquirement of chemically tailored heteroatom-doped GQDs via selective and quantitative C-N bond activation. In other words, the gist of this study is that, when using our method performed under a fixed process condition (170 °C, 10 min), GQDs with the desired chemical compositions can be obtained simply by changing the type of solvent without controlling the reaction conditions such as temperature and time.

In several previous reports on the preparation of heteroatom-doped GQDs from double precursors,^[5-9] it was mentioned that sufficient time is required for the functional moieties of N-containing dopants effectively to react with the terminal groups (e.g., carboxyl and hydroxyl groups) of GQDs.^[6,7] From a slightly different perspective,^[8,9] it was also reported that the synthesis of GQDs proceeds via two competition processes, consisting of molecular fluorophore-decoration on the surface of the QDs (namely, surface passivation) and carbon core growth. During this process, the relative rates between the formation of the molecular fluorophores and their consumption should be precisely controlled. To sum up, it means that eventually, when using conventional methods, there was no choice but to optimize the reaction conditions (e.g., reaction time, temperature, dopant ratio) to control the chemical configuration and optical properties of GQDs.

On the other hand, in our approach, the synthesis of NGs proceeds via two distinctive but simultaneous processes, consisting of edge-functionalization and carbonization. It means that there is no competition between surface passivation and carbon core growth, and consequently, there is no need to consider the effect of reaction conditions, including temperature, time, and concentration, on the chemical configuration of NGs.

Second, the synthesis of NGs proceeds via two distinctive but simultaneous processes, consisting of nitrile activation and carbon core growth. It means that if the reaction temperature or time is varied, the latter (core growth process) can also be inevitably affected. Although the optical features of our NGs are expected to be more strongly dependent on the presence of surface passivation groups (edge-functional groups) rather than on their size properties due to their large surface-to-volume ratio and complex surface states.^[10-13] in accordance with previous reports^[5] and our experience on the preparation of GQDs via a bottom-up approach from organic precursors, when the reaction temperature and time are excessively high or low, it is possible to obtain GQDs with several adverse features (e.g., a broad-size distribution, poor crystallinity, and low product yield). Therefore, in order to more accurately verify the effect of solvent on the structural features of NGs, as well as the relationship between the structural features and optical characteristics of NGs, other independent reaction parameters, including reaction time and temperature, must be strictly fixed at a specific value. Also, as mentioned above, if the reaction conditions are changed substantially, it is likely to obtain NGs with a broad size distribution, inevitably requiring the time-consuming size separation procedure. However, as mentioned in our manuscript, employing such post-treatments is rather against the objective of this research. From these reasons, we decided that the whole

experiments in this study are performed under one particular reaction condition (our condition: 170°C, 10 min).

From these perspectives, we and our coworkers did not consider verifying the reaction conditions (i.e., temperature, time) on both structural and optical characteristics of NGs by using experimental and computational methods. Also, it is expected that, even if the corresponding experiments are conducted, it will be difficult to exactly determine the origin of the change in structural and optical properties of NGs due to the combined effect of their chemical and morphological features.

In conclusion, as mentioned above, the purpose of this study that is based on the intermolecular interaction between FN and solvent molecules is not to suggest readers synthesize GQDs using the solvent presented in this paper, but to present a new strategy that researchers can employ to obtain GQDs with the desired chemical composition. Additionally, based on our preliminary result (Figure R5), it seems to be that the optical property of NGs is not significantly affected by the reaction conditions, in our system as long as it does not change substantially. (Of course, if necessary, we are willing to additionally conduct further experiments in order to offer appropriate responses to the reviewer.) **Therefore, we believe that the studies on the effect of reaction parameters on the properties of NGs prepared in a specific solvent system were outside of the scope in this study and were hence excluded.**

Figure R5. PL spectra of NG_Bu prepared under different reaction conditions (λ_{ex} : 400 nm).

(Comment #3_2) Will the change in reaction parameters lead to deviation between theoretical estimation and experimental results?

Response:

Thanks for the reviewer's consideration.

First, according to the formalism of DFT, its main purpose is not to investigate the time-dependent (dynamic) behavior of reaction system, but to get more deep insights into the physicochemical basis of host-guest chemistry in the system. Therefore, from this perspective, this comment seems to be not based on a general background in quantum chemistry.

Second, in thermodynamics, Gibbs free energy can be determined from $\Delta G_T = E_0 + ZPE + \delta H_T - T\delta S_T + E_{sol}$, where E_0 is the total electronic energy at the temperature of 0 K, ZPE is the zero-point vibrational energy of the molecule, δH_T and δS_T are the enthalpy and entropy changes from 0 to T K, respectively, and E_{sol} is the single solvation energy correction. In this equation, considering three boundary conditions ((1) $E_0 \gg ZPE + \delta H_T - T\delta S_T + E_{sol}$ & (2) $\delta H_T \gg \delta S_T$ & (3) no phase change in the material occurring during $C\equiv N$ activation) in our system, it is expected that there is little temperature dependence of Gibbs free energy change of this system.

Also, as mentioned above, owing to their large surface-to-volume ratio and complex surface states, the optical features of our NGs are more strongly dependent on the presence of surface passivation groups (edge-functional groups) rather than on their size properties.

Given these factors, if the reaction time or temperature is not excessively high or low, it is expected that the deviation between the theoretical estimation and the experimental results would not be high.

(Comment #4) This work showed a strong dependence between the optical properties and the types of edge-functional moieties. Previous studies also verified the effect of size distributions and N-bonding configurations on the optical properties graphene quantum dots. These should also be considered in this work.

Response:

Thanks for your consideration. You have raised a crucial point; however, we believe that the studies on the dependence of either size distribution or N-bonding configurations on the optical properties of NGs would not be only out of scope but also cannot provide meaningful information concerning their nature.

First, the main reason why optical characterizations, including modeling and spectrophotometric approaches, were performed in this study is to explore the correlation between optical properties and the end

moieties of GQDs (including the surface status of GQDs, which is passivated by the incomplete carbonization of organic precursors.). Also, as described in our manuscript, i) the average diameter and height of NGs are too small and ii) their distributions are quite narrow, and accordingly, they inevitably exhibit functionalization-dependent optical features rather than size-dependent absorption and PL characteristics (e.g., conjugation-elongation effects). Moreover, as mentioned above, due to their large surface-to-volume ratio and complex surface states, the optical features of our NGs are more strongly dependent on the presence of surface passivation groups (edge-functional groups) rather than on their size properties. From these perspectives, during the preparation of this manuscript, we and our coworkers did not consider verifying the effect of size distributions on the optical properties of GQDs by using experimental and computational methods.

Second, our group^[14] and other researchers^[15] were previously reported that three types of substitutional N species (e.g., quaternary N (N_q), pyridonic N (N_{pydo}) and pyridinic N (N_{pydi})) hardly affect the optical properties of hexagonal GQDs (Figure R6), because certain *s*-orbital contributions to the frontier energy levels are not observed in these structures. Instead, according to our results (Figure 4b in manuscript) and related studies, substitutional N-doping in GQDs can lead to the improvement of either their electrical conductivity or catalytic activity. **From this point of view**, the additional experiments and computational studies on the effect of N-bonding configurations on the optical properties of GQDs by using the TD-DFT methods with B3LYP and CAM-B3LYP functionals were not performed in this manuscript.

Figure R6. (a) Theoretically simulated PL spectra of C₇-GQDs with different types of edge-functional groups. (b) Theoretically simulated PL spectra of C₇-GQDs with different N-configurations.

Reference

1. Moon *et al.*, *Chemistry of Materials* **2016**, 28, 1481-1488
2. Zhang *et al.*, *Journal of Catalysis* **1999**, 182, 70-81.
3. Smirniotis *et al.*, *Applied Catalysis A: General* **1999**, 176, 63-73.
4. Badii *et al.*, *Polymer Degradation and Stability* **2016**, 131, 53-61.
5. Qu *et al.*, *Scientific Reports* **2014**, 4, 5294.
6. Qu *et al.*, *Nanoscale* **2013**, 5, 12272-12277.
7. Dong *et al.*, *Carbon* **2012**, 50, 4738-4743.
8. Song *et al.*, *Journal of Materials Chemistry C* **2015**, 3, 5976-5984.
9. Tang *et al.*, *Sensors and Actuators B: Chemical* **2018**, 258, 637-647.
10. Hola *et al.*, *Carbon* **2014**, 70, 279-286.
11. Tang *et al.*, *ACS NANO* **2012**, 6, 5102-5110.
12. Xu *et al.*, *ACS NANO* **2013**, 7, 10654-10661.
13. Sun *et al.*, *Journal of the American Chemical Society* **2006**, 128, 7756-7757.
14. Moon *et al.*, *Nano Energy* **2017**, 34, 36-46.
15. Lin *et al.*, *Journal of Computational Chemistry* **2018**, 39, 1387-1397.

Reply to Reviewer #2

The paper describes a solvents mediated growth of graphene quantum dots using a single source precursor procedure integrating some experimental evidences with a stronger theoretical calculation. I believe that the work will positively contributes to active and practical utilization of graphene based nanomaterials in several applications. I recommend accepting the work but I would suggest authors to make a major revision by the following comments and considerations.

(Comment #1_1) The chemical approach described in this paper is not substantially different for the experimental procedure reported elsewhere [Byung Joon Moon.... and Sukang Bae, Chem Mater 2016, 28, 1481-1488] where the synthesis of nitrogenated graphitic carbon dots has been described (“..0.3g FN dispersed in 10 mL DI water was injected into a 50 mL round bottle flask, and then heated to 225°C for 10 min”...while in the current paper... “0.1 g of FN was dispersed in 10 mL of each solvent injected into a 100 mL Teflon liner, and then heated to 170°C for 10 min”..). Considering that the differences observed in the current/submitted paper are from authors ascribed to supporting solvent molecules (SSM effect) that act as pseudo-electron-withdrawing moieties for C≡N group and/while the primary precursor (FN) activation is assumed to be supplied from air moisture within the reaction chamber, how do they think that the described synthetic mechanism would be influenced by different reaction conditions such as an increasing the concentration of FN and/or an increasing the reaction temperature?

Response:

Thanks for the reviewer’s kind consideration.

First of all, we believe that it is difficult to compare the two studies (our previous work^[1] vs present work) directly in terms of either absolute temperature or concentration because each experiment was performed using different reaction systems. The main reason for changing the reaction system (semi-closed conventional reflux batch reactor (50 mL round bottle flask) → fully closed-autoclave reactor) was to enable large-scale synthesis by breaking away from the restriction of reactor size. Also, owing to that the reaction vessel is not to be exposed to the surrounding atmosphere, this reaction system can offer rapid and uniform heat energy to a reaction

medium, allowing the reaction temperature (or time) to be somewhat reduced.

Second, the synthesis of NGs proceeds via two distinctive but simultaneous processes, consisting of nitrile activation and carbon core growth. It means that if the reaction temperature or time is varied, the latter (core growth process) can also be inevitably affected. Following previous studies,^[2-5] due to their large surface-to-volume ratio and complex surface states, the optical features of our NGs are expected to be more strongly dependent on the presence of surface passivation groups (edge-functional groups) rather than on their size properties. Nonetheless, based on previous reports^[6] and our experience on the preparation of GQDs via a bottom-up approach from organic precursors, when the reaction temperature and time are excessively high or low, it is expected that the optical properties of GQDs will vary due to their broad-size distribution and poor crystallinity.

Therefore, in order to more accurately verify the effect of solvent on the structural features of NGs, as well as the relationship between the structural features and optical characteristics of NGs, other independent reaction parameters, including reaction time and temperature, must be strictly fixed at a specific value. For this reason, we decided that the whole experiments in this study are performed under one particular reaction condition (our condition: 170°C, 10 min). Also, we believe that the studies on the dependence of either reaction temperature or time on the optical or structural properties of NGs would be out of scope for this study because, in this paper, we focused on the availability of this platform to be used as a predictable synthetic tool for the acquirement of chemically tailored heteroatom-doped GQDs via selective and quantitative C-N bond activation. In other words, the gist of this study is that, when using our method performed under a fixed process condition (170 °C, 10 min), GQDs with the desired chemical compositions can be obtained simply by changing the type of solvent without controlling the reaction conditions such as temperature and time.

Third, in our previous study, we already found that each GQDs prepared in the concentration range of 10-30 mg/mL exhibited similar chemical compositions, as well as optical properties. Also, in this paper, the primary purpose of changing the concentration of FN was to ensure both solution and reaction homogeneities during the synthesis process of NGs under different solvent systems.

To sum up, we believe that the studies on the dependence of either temperature or concentration on the structural and optical properties of NGs would either be out of scope for this study or not provide meaningful information concerning their nature. Nevertheless, as requested by the reviewer, we are willing to partially conduct

further experiments to offer appropriate responses to the reviewer.

(Comment #1_2) In other words, which is the main reason that allows the production of graphene and graphite with the two different while similar synthetic procedures since the FN overall concentration is not extremely different within the two procedures?

Response:

Thank you for the reviewer's kind consideration. First of all, we and our coworkers apologize for any inconvenience/confusion this may cause you and the reviewers. In actuality, the products (N-GCDs) synthesized using the previous method^[1] were also graphene quantum dots, not graphite quantum dots. Our samples were initially called "nitrogen-doped graphene quantum dots" (N-GQDs), but at a reviewer's request, they were renamed "nitrogen-doped graphitic carbon dots" (N-GCDs). That is to say, both products differ only in their names and are characteristically identical.

(Comment #1_3) If they would consider the synthetic temperature as the main factor that would preferentially promote a bidimensional (1-2 layered NGs) or "three dimensional" (4 layered NGs) as described in Scheme S1, do they have any experimental result about the here reported nitrogenated graphene quantum dots synthesis in presence of benzaldehyde, but heating the system at 225°C for 10 minutes?

Response:

First, as mentioned above, it is difficult to compare the two studies (previous work^[1] vs present work) directly in terms of either absolute temperature or concentration because each experiment was performed using different reaction systems (reflux batch reactor vs autoclave reactor). Also, as mentioned in our manuscript and supplementary information, more analyses of the exact origin of the morphological variation are ongoing and will be included in our future studies. Until now, as mentioned in the supplementary information, it has been ascribed that the formation of stitched graphitic sheets is dependent on the variety of reaction kinetics, such as the 1) steric hindrance of functional groups in multimeric forms, 2) the molecular geometry of intermediate molecules, and 3)

the reaction selectivity for the formation of electronically (or geometrically) stable graphitic nanostructures. That is, given this complexity, it is evident that such a phenomenon cannot be determined simply by the reaction temperature.

Also, according to reviewer's comment, we attempted to synthesize NG_Be under previous reaction conditions, and acquired its AFM image as shown in Figure R7. Given this, it can be confirmed that the average height of NG_Be is likely to be influenced a little by the reaction temperature, but the difference between two samples is not enough to consider the synthesis temperature as the main factor determining the growth direction of NGs.

Figure R7. AFM height images showing the corresponding height profiles of NG_Be (condition: 225°C, 10 min).

(Comment #2_1) In the abstract and into the introduction authors describe their procedure as scalable (line 27) and self-assembled reaction (line 92). I haven't find any concrete indication, evidence or purely explanation in the paper that the procedure is effectively scalable at larger quantity reaction batches.

Response:

Thanks for the reviewer's kind consideration. We and our co-workers apologize in advance for any inconvenience/confusion those omissions may cause you and the reviewers.

As described in our manuscript, despite the early successes in the production of heteroatom-doped GQDs, there still exist many challenges in the optimization of fabrication processes, including the removal of acidic/basic intermediates and dopant residues, which necessarily involve time/cost-consuming post-treatments and hinder large-scale manufacturing. Besides, when using conventional techniques, there are many of obstacles (e.g., amorphous structure of dopants, high volatility of dopants, and uncontrollable kinetics of subsequent

competitive reactions) that make it difficult to simultaneously control both the structural features (crystallinity) and doping characteristics (concentrations and configurations).

In contrast, our strategy readily accesses to the qualitative and quantitative control of the GQD composition. In addition, it does not involve time/cost-consuming post-treatments for the removal of residual reactants,^[1] such as doping additives and molecular fluorophores like-byproducts, and thus large-scale manufacturing for industrial application is possible. Unlike the submitted manuscript, its earlier version had included some statements (in supporting information), along with corresponding evidence as shown below (Figure R8), on the advantage of our approach for scalable production of N-doped GQDs.

Figure R8. (a) Photograph of apparatus for large-volume preparation of NGs. (b) Photograph of NG_DW dissolved in ethanol. (c) XPS survey and (d) PL spectrum of NG_DW (reference spectrum displayed in Figure S20, λ_{ex} : 370 nm).

(Complemented supporting information)

Figure S5. (a) Photograph of apparatus for large-volume preparation of NGs. (b) Photograph of NG_DW dissolved in ethanol. (c) XPS survey and (d) PL spectrum of NG_DW (reference spectrum displayed in Figure S20).

(Comment #2_2) Moreover, the self-assembly regards a specific concept concerning the formation of highly ordered structures, in which, ideally, diffraction methods allow one to deduce structural periodicities to quantify the order beyond a coarse microscopic observation. My feeling the self-assembly of FN is not the correct way to describe the system and is not concretely supported in the paper, whereas the reaction is more a thermolytic polymerization rather than a chemical self-assembly.

Response:

Thanks for the reviewer's kind consideration. We agree with the reviewer's comment that our reaction is considered to be one of the thermolytic polymerization reactions. Strictly speaking, as mentioned by the reviewer, ideal self-assembly is defined as the spontaneous organization of molecules under thermodynamic equilibrium conditions into structurally well-defined and rather stable arrangements, leading to the structural periodicities that are enough to be measured by employing diffraction or high-resolution microscopic technique. However, such a term "self-assembly" is also used in the more general sense of growth of GQDs via a bottom-up approach from organic precursors. Specifically, when the sample is heated under hydrothermal conditions, through spontaneous intermolecular dehydration (or deamidation) between the carboxyl, hydroxyl, and amino groups of intermediates, they gradually grow into nanosheet structures, which is often referred to as a "self-assembly reaction" in the field of synthesis of GQDs via bottom-up approaches.^[7-9] In other words, it means that the formation of GQDs via carbonization process can be described as the self-assembly of intermediates, which is based on molecular recognition (via H-bonds, electrostatic interactions) between those with various O- and N-containing groups.

From this perspective, we and our co-workers ask for your reconsideration. And, if you still feel that the term, "self-assembly", is not suitable to describe our system, we are willing to replace it with "thermolytic polymerization (your suggestion)".

(Comment #3) The oxidative activation of C≡N group is activated by moisture. Is there any specific reason or explanation that allows the partial oxidation of only one C≡N group of FN molecule and not both C≡N

groups?

Response:

Thank you for your careful consideration. We believe that you have ask an interesting question. As described below, there are two reasons why the partial oxidation model was chosen to trace the development of chemical reactions in each solvent.

First, considering the steric and electronic effects in nucleophilic substitution reactions, it is expected that a reaction in which one of the two C≡N groups is preferentially activated would be energetically preferred over the simultaneous oxidation of both C≡N groups.

Second, in mechanism studies, it is extremely hard to take all components into consideration since the real systems are complex and need to perform a large number of calculations; hence, the selection of appropriate models is inevitably required. In this study, we proposed that selective and quantitative C-N bond activation of FN is fundamentally based on its intermolecular interaction with solvent molecules. Therefore, to validate such mechanism cost-efficiently yet accurately, we selected a simplified but representative model (e.g., partial oxidation process of C≡N group) for our reaction system. Considering our reaction system, at any one particular moment, it is expected that a reaction in which one of the two C≡N groups is preferentially activated would be stochastically preferred over the simultaneous oxidation of both C≡N groups. Of course, even considering the preliminary results obtained using the simultaneous oxidation model, the underlying mechanism of intermolecular interaction between FN and solvent molecules was sufficiently verified.

(Comment #4) In the main text of the paper, are often reported some abbreviations not previously described that makes the reading uncomfortable.

- a. Line 148: Fam that I believe is fumaroaldehyde-like molecule, but it is not explicitly announced in the text.
- b. Line 175 and Fig1f, acronyms as IMs, TR1, IM6, TR2, etc..that I believe to be reaction intermediates, but are not at all described or explained, I think it makes the paper not clearly understandable from a reader.

Response:

Thank for appropriate comment. We also agree on the reviewer's comment entirely. Following

reviewers' comments, we have corrected our manuscript as below.

(Corrected manuscript)

1) [page 8, line 3] First, the two most plausible mechanisms for the C-N activation of FN (FN → FAm (fumaramide-like molecule)) are depicted at the top of Figure 1d, where Paths I and II represent the general solvolysis reaction and solvent-catalysed oxidation reaction, respectively.

2) [page 8, line 26] In particular, it is clearly observed that there is a distinct difference in the ΔG value of the rate-determining region (intermediate state structure #2 (IM2) ~ transition state structure #1 (TR1) & IM6 ~ TR2) in each reaction step.

(Comment #5) (#Q1) Few figures reported in the supporting material are not even mentioned in the main text, such as Fig S1 (Raman) and Fig S2 (NMR) which might contain in principle a more detailed information but are not used in the main text, making them useless. (#Q2) Fig S3, which is only noise or a not-emissive sample, should be integrated of numerical counts on y axis in the way of making clearer that represent a “zero” value.

Response:

We thank the reviewer for the careful consideration. First, we agree to a certain extent with the reviewer's comment #Q1. Indeed, we were concerned about it in preparing our manuscript. Nevertheless, as you perceived, since our manuscript already contained too much figure and description, we have inevitably included detailed contents on the synthesis mechanism of NGs entirely in the supplementary information. So please we ask for your understanding about this matter.

Second, we agree with the reviewer's comment #Q2 entirely. According to the reviewer's comment “Q2”, we have corrected Figure S3 as below.

(Corrected supporting information)

Figure S3. PL spectrum of the intermediate product with relatively high oxidation state of FN (λ_{ex} : 375 nm).

Reference

1. Moon *et al.*, *Chemistry of Materials* **2016**, 28, 1481-1488.
2. Hola *et al.*, *Carbon* **2014**, 70, 279-286.
3. Tang *et al.*, *ACS NANO* **2012**, 6, 5102-5110.
4. Xu *et al.*, *ACS NANO* **2013**, 7, 10654-10661.
5. Sun *et al.*, *Journal of the American Chemical Society* **2006**, 128, 7756-7757.
6. Qu *et al.*, *Scientific Reports* **2014**, 4, 5294.
7. Taspika *et al.*, *RSC Advances* **2019**, 9, 7375-7381.
8. Hussain *et al.*, *Journal of Molecular Liquids* **2020**, 318, 114052.
9. Fresco-Cala *et al.*, *RSC Advances* **2018**, 8, 29939-29946.

REVIEWERS' COMMENTS

Reviewer #1 (Remarks to the Author):

The authors have revised the manuscript well according reviewers'suggestion.Now,I agree its publication in its present form.

Reviewer #2 (Remarks to the Author):

I am still not convinced that this paper is suitable for being published in Nature Communication